# Adolescent Knowledge, Attitudes and Practices of Healthy Eating: Findings of Qualitative Interviews among Hong Kong Families

**DOI:** 10.3390/nu14142857

**Published:** 2022-07-12

**Authors:** Kiki S. N. Liu, Julie Y. Chen, Kai-Sing Sun, Joyce P. Y. Tsang, Patrick Ip, Cindy L. K. Lam

**Affiliations:** 1Department of Family Medicine and Primary Care, The University of Hong Kong, Hong Kong SAR, China; snliu@connect.hku.hk (K.S.N.L.); joycetpy@hku.hk (J.P.Y.T.); clklam@hku.hk (C.L.K.L.); 2Department of Family Medicine, The University of Hong Kong Shenzhen Hospital, Shenzhen 518053, China; 3JC School of Public Health and Primary Care, The Chinese University of Hong Kong, Hong Kong SAR, China; tonysun@cuhk.edu.hk; 4Department of Paediatrics & Adolescent Medicine, The University of Hong Kong, Hong Kong SAR, China; patricip@hku.hk

**Keywords:** healthy eating, adolescents, knowledge, attitudes, practices, qualitative

## Abstract

To tackle unhealthy eating among adolescents, it is crucial to understand the dietary knowledge, attitudes, and practices (KAP) on which adolescent eating habits are based. This qualitative study identifies the gaps in KAP by exploring what Chinese adolescents know, perceive, and practice regarding healthy eating to better inform targeted interventions for this important health problem. Parent–adolescent dyads were purposively sampled based on, for example, the dietary intake, age, and gender of the adolescent and household income, and each completed a 30 to 60 min interview. Twelve themes were synthesized: knowledge: (1) dietary recommendations, (2) health outcomes of healthy eating, (3) nutrition content in food, and (4) access to healthy meals; attitudes: (5) outcome expectation for healthy eating, (6) food preferences, and (7) self-efficacy regarding adopting healthy eating; and practices: (8) going grocery shopping for healthy food, (9) eating home-prepared meals. (10) eating out in restaurants or consuming takeaway food, (11) fruit and vegetable consumption, and (12) snacking, perceived unhealthy eating to be low risk, made unhealthy choices regarding snacking and eating out, and had insufficient fruit and vegetable intake. Programs should emphasize the positive short-term health outcomes of healthy eating and empower adolescents to acquire food preparation skills to sustain healthy eating habits.

## 1. Introduction

Non-communicable diseases (NCDs) are the global leading cause of death [1], with 22% of global deaths among adults being attributed to diet-related NCDs—cardiovascular diseases, cancers, and diabetes mellitus [2]. Nutrition transition to a reduced intake of fiber-rich food and increased intake of convenience food high in fat, salt, and sugar contributes to increased NCD risk [2,3,4]. The eating habits of the majority of adolescents are substandard to healthy dietary recommendations [5,6,7]; this may explain the growing prevalence of adolescent obesity [8,9], which increases the risk of NCD in adulthood [10,11].

Knowledge, attitudes, and practices (KAP) are the key elements of promoting change in the behavior of an individual. They work together by providing cognitive understanding, forming beliefs and attitudes, and initiating attempts to achieve behavioral change. They are crucial to understanding the reasons for unhealthy eating among adolescents, because adolescents develop personal eating habits based on their dietary KAP. Identifying the gaps in their KAP can therefore help us to design more effective targeted interventions.

There is a broad spectrum of knowledge that needs to be acquired with respect to healthy eating, such as knowledge on dietary guidelines, health outcomes, and preparation methods. Health authorities have dietary guidelines for the recommended intake for each food group. In Hong Kong, for example, the government has been promoting “Two Plus Three” for the recommended daily servings of fruit and vegetables (FV) [12]. It is also important to know about the health outcomes resulting from eating habits—such as body weight control, gastrointestinal health, and the risk of developing chronic diseases such as diabetes and cardiovascular diseases—as well as food choices and preparation methods, such as how to interpret nutrition labels and healthiness of various cooking methods. Adolescents should acquire nutritional knowledge as part of their school curriculum [13], though this would only cover foundational content. A comprehensive exploration of all aspects of food knowledge among adolescents is lacking, which may explain the inconsistent results obtained regarding the association between dietary knowledge and habits across studies [14].

Knowledge enhances positive attitudes in terms of the perceived health consequences of unhealthy eating, the efficacy of eating healthily [13], and food preferences among adolescents [15]. Although many adolescents, with numbers as high as 87% to 94.2% observed in previous studies [13,16], report healthy eating as being important, their food choice is affected by many other factors [17]. Flavor and cost often compete with health considerations when adolescents and young adults select food items [13,18]. A qualitative study conducted in Hong Kong found that the unhealthy eating habits of adolescents from low-income families were related to cost and their taste preference for unhealthy food [19]. Further exploration of the tensions existing among health, taste, and other preferences when making food choices is warranted.

Food practices include food purchasing, meal patterns, and eating habits, and healthy practices help people to meet their dietary recommendations. Apart from a sufficient FV intake of at least five servings daily, healthy eating habits should limit the intake of fat, sugar, and salt from commonly consumed foods, such as instant noodles, processed foods, confectionaries, and soft drinks [20]. These habits should also involve choosing healthier cooking methods for both home and restaurant meals. Local health authorities have suggested some tips specific to healthy cooking, eating out, and snacking. Examples of healthy habits include using low-fat cooking methods, such as steaming and braising, choosing natural herbs and spices as seasonings for home-prepared meals [21], asking for separate serving of sauces and sugar when eating out, snacking no more than once between two main meals, and selecting natural fresh foods as snacks [22]. While eating at home is a common practice in Hong Kong, the majority of adolescents consume insufficient FV [7], probably due to the cost and their insufficient nutritional knowledge [21]. They also report excessive salt and sugar intake [5,6,7]. Another area of concern in this regard is the practice of frequent snacking among adolescents, with large proportions reporting a daily snacking practice in China [23] and a twice-daily snacking habit in Malaysia [24]. Despite the availability of healthy choices, most consume snacks that contain empty calories [24,25].

Eating culture and cooking methods are region-specific, and only a few qualitative studies related to healthy eating among adolescents in Hong Kong have been conducted. In the qualitative study by Chan et al., all twenty-two students aged 13–15 years included had a general concept of healthy eating regarding the recommended servings of FV and examples of foods and eating habits that were healthy and unhealthy [26]. Their attitudes towards healthy eating, however, were not examined. Siu et al. explored the difficulties specifically encountered by low-income families and identified the low priority given to health when compared with price and taste when eating out [19]. This qualitative study therefore aims to provide an in-depth comprehensive exploration of the knowledge, attitudes, and practices relating to healthy eating among adolescents in order to identify the gaps in KAP to inform public health strategies.

## 2. Materials and Methods

### 2.1. Subjects

The sampling population was families who had joined a previous health assessment program of another study, the Trekkers Family Enhancement Scheme (TFES) and comparative family cohort study [27], which targeted low-income families and provided nutrition and healthy cooking workshops to some TFES families. We identified 288 families with an adolescent aged 10–19 years old by 31 December 2019, in which the adolescent also completed a dietary intake survey. Families with a parent or child not able to speak Cantonese or in which neither parent was the primary caregiver of the child were excluded.

While the sample size for qualitative research is determined by data saturation [28], three previous qualitative studies on the same topic with parent–child paired interviews [29,30,31] included 15 to 26 families. Based on this, we planned to first recruit 20 families and to recruit more as necessary until data saturation was reached. The interviews and coding were carried out concurrently. Data saturation was deemed to have been achieved when no new codes were generated from the interview transcripts after 3 consecutive interviews.

We applied stratified purposive sampling to select families whose adolescents had “Healthy”, “Average”, and “Unhealthy” eating habits based on the results of the self-reported dietary intake survey. Adolescents with a daily consumption of 5 or more servings of FV were classified as healthy, 3–4 servings as average, and 2 or fewer servings as unhealthy. We also sampled families with adolescents of different ages and genders, families with household income below and above the population median, and families who had attended previous nutrition workshops. The sampling strategy aimed to enrich the data by capturing variation and depth of perspectives from different families and to explore the possible influence of family income and previous training.

Parents in the selected families were consecutively invited by phone to participate in the study until the required number of families had been recruited. Written consent was collected from the parents and their adolescents before the interview. A total of 136 families (31 with healthy adolescents) were contacted between May 2020 and March 2022. Of these, 22 families agreed to participate in the study and 3 additional families who joined TFES in 2021 were recruited.

### 2.2. Interview Setting

The aim and details of research were explained to the participants before the interview. All the family interviews were conducted by a member of the research team (K.S.N.L.), an experienced qualitative researcher, with a trained research assistant as an observer who took field notes. The interviews were carried out on Zoom with the camera on, except for with one family who refused.

### 2.3. Data Collection

A KAP framework was constructed incorporating four constructs adapted from social cognitive models—namely, (1) outcome expectations (beliefs in the health benefits of healthy eating); (2) subjective norms (acceptance by family/peers); (3) self-efficacy (in complying to dietary guidelines); and (4) cues to action (personal food environments). The framework covers “Knowledge on dietary guidelines, health benefits/harms, healthy food choice and access; Attitudes in believing, accepting and complying with dietary guidelines for health benefits; and Practices of personal and family eating behaviors, including home food environment and family meals” (Figure 1). A similar framework was shown to be applicable to exploring adolescents’ eating habits [32]. A semi-structured interview guide (Appendix A) was developed based on the theoretical framework and the guidelines established by the Food and Agriculture Organization of the United Nations [33]. It aided in collecting the families’ views on the facilitators and barriers of knowledge, attitudes, and practices of adolescent healthy eating. Potential strategies for resolving the barriers and promoting the facilitators were also explored. We field-tested the dyad interview format and data collection method on two families to confirm its feasibility and to further refine the interview guide with the addition of sub-questions. To engage the adolescents and encourage them to express their views, we first directed the questions to them and observed the verbal and non-verbal communication between the adolescents and their parents.

The interviews were conducted in Cantonese and audiotaped. Each interview lasted 30 to 60 min. Data saturation was reached after 25 interviews, at which point no new findings emerged in 3 consecutive interviews.

### 2.4. Data Analysis

We used thematic analysis to analyze and report the results by theme identification [34], which is “independent of theory and epistemology” and allowed us the flexibility to address most of the research questions [34]. To ensure the credibility of the result, a 6-phase guide was followed for the analytical process—(1) familiarizing ourselves with the data; (2) generating initial codes; (3) searching for themes; (4) reviewing themes; (5) defining themes; and (6) completing write-up [34,35].

The recordings of the interviews were transcribed verbatim in Chinese. The NVivo software was used to carry out the data analysis. A coding tree on the hypothesized themes and sub-themes based on the KAP framework was constructed to facilitate data extraction from the transcripts. The data of each transcript were coded independently by two research assistants according to the coding tree. Additional themes added to the coding tree as new factors were identified from the transcripts. The coding consistency between the two coders was checked and inconsistencies were resolved by discussion among the coders and authors to reach agreement. The findings on adolescent KAP were categorized into those areas for which general agreement was reached among the families and those in which consensus regarding KAP was lacking. These KAP gaps were areas of healthy eating which the adolescents were insufficiently aware of, had misconceptions about, or failed to achieve. Selected quotes relating to the key findings were translated into English for reporting.

## 3. Results

### 3.1. Subject Characteristics

Twenty-five adolescents aged 12 to 19 years and their mothers aged 40 to 56 participated in the study. The distribution of each characteristic is shown in Table 1, and Table 2 lists the subjects by family dyad. Thirteen of the adolescents were female, and eight reported meeting the healthy eating recommendation of consuming 5 or more FV servings per day in a dietary intake survey or during the interview. Twenty-one families had a household income below the population median (HK$27,000), and eight of them were below the poverty threshold of 50% of the population median (HK$13,500). Ten parents had participated in previous nutrition workshops held by the TFES before the interviews.

### 3.2. Adolescent KAP of Healthy Eating

Twelve themes were synthesized from the interviews and categorized under the KAP theoretical framework. The Appendix A provide more quotes supporting the findings obtained for each theme.

#### 3.2.1. Adolescent Knowledge of Healthy Eating

There were four themes under knowledge construct (Table 3).

Dietary recommendations

Relative portions of food categories—The knowledge on dietary recommendations among the adolescents focused on the relative portions of food categories. They knew about the food pyramid and that consuming more vegetables and less sugar, oil, and salt aided healthy eating.

“Always those ‘more vegetables, less meat’, usually those classifications... that means do not only eat that much meat.”(A8, M, 14)

Recommended daily servings or allowance—When asked specifically about the recommended servings of or allowances for the food categories, only a few adolescents were able to correctly mention two servings of fruit and three servings of vegetables as well as grains, vegetables, and meat in a ratio of 3:2:1 in a lunchbox. Most were unaware of the exact intake limits for unhealthy food categories such as salt and sugar. One respondent gave an example of food with excessive salt content, although she was unsure of the quantity.

“Have heard of the salt content in [the noodle soup of] the Tam’s [Yunnan Rice Noodles] may exceed... the daily limit for a person.” (A1, F, 18)

Underestimating the recommended servings—Some misinterpreted the lunchbox portion guideline and had the wrong idea about the recommended servings of FV, which was usually underestimated by those with insufficient intake.

“Do have one bowl [of vegetables] indeed... Enough [consumption]… Eating [daily] is fine already.”(A18, M, 19)

2.Health outcomes of healthy eating

Observable short-term outcomes—Although the adolescents were able to associate healthy eating with general health and disease prevention, they seemed to focus only on the short-term outcomes and neglected to consider the longer-term outcomes of healthy eating. The commonly cited health consequences tended to be those that were directly observable, such as constipation, weight gain, and acne.

“Eating too oily will have acnes... may have throat pain.” (A7, F, 14)

Specific benefits of eating FV—Some of those who consumed sufficient FV highlighted a few less well-known benefits of eating FV—for example, for the relief of body pain and improved skin condition.

“I think the major benefit is having smooth excretion, less likely to have constipation. And skin condition will be better if eating more FV, having less severe acnes and keeping moisture in skin.” (A1, F, 18)

Long-term outcomes—Only a few adolescents mentioned the long-term health outcomes of eating habits, such as the risk of chronic diseases and tissue repair by vitamins.

“[Eating more FV] is less likely to have constipation or cancer, and fruit is rich in vitamin C. It can help us repair the areas that are often injured... These are learnt from class.” (A12, F, 18)

3.Nutrition content in food

Food sources of nutrients—The adolescents knew about a variety of unhealthy foods that were high in fat, salt, and sugar, such as salty snacks, confectionaries, and restaurant food. Some identified the types of food that contained specific nutrients, such as protein and vitamins, that were beneficial for body strength and function.

“Those vegetables are good for eyes, those carrots... vitamins... may be beneficial to the eyes. Or blueberry, I will eat.” (A24, F, 17)

Interpretation of nutrition label or claims—On the other hand, many adolescents had learnt how to read nutrition labels. Those who followed healthy eating habits used the nutrition label to make healthier food choices, such as looking at the sugar content.

“Usually read the sugar content when buying [sugar drinks]… I usually buy those with around 5 g per 100 mL… like sugar-free tea.” (A11, F, 13)

Healthy snack options—Most could not provide examples of healthy snacking choices. Some mentioned a few healthy options, such as yogurt, fruit, and nuts.

4.Access to healthy meals

Cooking methods—The adolescents knew that eating out or eating takeaway food was unhealthy because these establishments tend to use unhealthier cooking methods with the excessive addition of seasonings and oil.

“I think [take-away] is less healthy, maybe there are oily foods… and the drinks are very sweet, and may not know what seasonings are used, may add much oil or salt, so may not be as healthy as home [meals].” (A7, F, 14)

Ways to identify healthier restaurant meals—Some adolescents knew about healthier cooking methods—for example, steaming and boiling. However, most lacked knowledge about making healthy choices in restaurants, except for a few, who mentioned smaller meal portions and the provision of vegetables.

“If [I] want to eat more healthily for rice noodles, [I] will reduce the portion of noodles, and order an extra dish of vegetables.” (A20, F, 17)

#### 3.2.2. Adolescent Attitudes towards Healthy Eating

There were three themes under attitude construct (Table 4).

5.Outcome expectation for healthy eating

Healthy eating is necessary based on experience—Many perceived that healthy eating was necessary for maintaining health and weight control. Some were able to relate to their previous experiences, most of which were negative outcomes from unhealthy eating.

“Easy to get diarrhea when eating [foods that are] too oily, because my stomach is not that resistible.” (A25, M, 18)

Two respondents described their positive experience with having regular bowel movements and feeling energized after eating more FV.

“Having less severe constipation [after eating vegetables]… Have experienced.” (A9, F, 17)

“I will replace the carbohydrates from starch by vegetables and fruit, so feel like less easy to get sleepy.” (A20, F, 17)

Being only necessary for older adults—Some adolescents perceived themselves as having low susceptibility to the harm caused by unhealthy eating because they were young and thus had a better metabolism. They thought that healthy eating was only necessary for older adults and not for themselves.

“I personally think that if... before reaching a certain age... maybe unnecessary to reduce eating oily food, but not eating too much... It is acceptable to eat certain amount, because of the metabolism, isn’t it? Can digest the food very quickly.” (A24, F, 13)

6.Food preferences

Taste preference for unhealthy food—The adolescents chose their food based on several preferences. The most frequently mentioned was taste, especially when they ate out or snacked. Most of the choices made based on taste were unhealthy and included fried food, savory street food, and confectionary.

“Will eat again if [I] think [it is] tasty… Those from Bafang [Dumpling]… the flavor is heavier, usually those fried food smells better.” (A19, F, 12)

Taste perception of healthy food—While some adolescents liked the taste of FV and low-sugar drinks, others expressed dislike towards healthier options of snacks and certain vegetables due to their inferior taste.

“Healthy [snacks] are usually not tasty, so not buying [them].” (A5, M, 15)

Priority of different preferences—A few prioritized health over taste when choosing food items or cooking methods. Others emphasized that they were unwilling to adopt healthy eating practices because they had other priorities, such as taste and convenience, over health.

“Have learnt about [the dietary recommendations], but not following… Lazy to count the servings.” (A6, F, 14)

The adolescents’ autonomy around and unwillingness to choose healthy food options were also observed by their parents.

“I know [soft drinks are] bad for health, but [the children] also know [these are] not good for health, sometimes they still drink.” (P3, F, 55)

7.Self-efficacy regarding adopting healthy eating

Assessing health by body shape—Most adolescents perceived that they had self-efficacy to follow a healthy diet if they wanted to. Some explained that their eating habits were determined by the perception of their body shape and that they control the amount they consume, particularly with regarding to unhealthy food.

“You will not eat that much when you see the portion is too large. Honestly, you assess how much you can absorb, but not rigid for everything… Indeed we will do exercise, then there will be a balance.” (A18, M, 19)

Strategies to eat healthily—Some illustrated that they were able to eat more healthily with friends or on their own by cooking themselves and making better restaurant choices.

“Used to [eat fast food] more in the past, more restricted now... Going less to the fast food shops [even with friends].” (A17, M, 15)

Lack of food preparation skills—Two adolescents admitted that they did not know how to prepare FV, since this was the sole responsibility of their parents. This was a barrier to them consuming FV on their own.

“I do not know how to buy [or choose FV], not to mention that not knowing how to cook… Limited time [to learn from mum].” (A14, F, 13)

#### 3.2.3. Adolescent Practices of Healthy Eating

There were five themes under practice construct (Table 5).

8.Grocery shopping for healthy food

Choice vs. quantity—Most adolescents did not engage in grocery shopping for food. They occasionally purchased snacks on their own and seldom read nutrition labels due to their preference for taste and convenience. However, they might control the quantity they ate instead of choosing healthy alternatives.

“Not reading [nutrition label] intentionally... because it feels like everything needs to be counted. [I] may prefer eating if want to, may eat less.” (A24, F, 13)

Reading nutrition labels to determine healthy alternatives—A few female adolescents were quite conscientious of shopping for healthy foods and checked the nutritional label of beverages for their sugar content before purchasing.

“Only read sugar and fat, and total calorie… If the sugar content is over 3 [gram], I mostly not drinking.” (A12, F, 18)

Grocery shopping with parents—Some adolescents would sometimes accompany their parents to the supermarket, who often would share their practice of reading nutrition labels.

“[Reading nutrition label] is done habitually. Sometimes we will read together when she accompanies me to the supermarket. For example, we like certain brand of biscuit, and will compare the sugar, fat or whatever [on the label content] among different flavors before making a choice.” (P1, F, 51)

9.Eating home-prepared meals

Quick breakfast prepared by parents—The adolescents usually had their meals at home. Most of them were not involved in meal planning nor preparation, which was seen as the parents’ responsibility. They had a quick breakfast in the morning before going to school. Parents tended to buy ready-to-eat or prepare easy-to-cook food, such as ham, buns with fillings, and processed meatballs, which may or may not be healthy.

“Preparing sandwiches, with egg and ham… sometimes with cheese… Steamed vegetable and meat bun, barbecue pork bun, sometimes with red bean, sesame paste.” (P4, F, 56)

Balanced meals vs. self-cooking—For lunch and dinner, the meals prepared at home were usually more balanced, including grains, vegetables, and meat. When the adolescents cooked their own meals, they preferred easy-to-cook options—e.g., instant noodles and processed meat—which were mostly unhealthy.

“I will cook if I wanna eat instant noodles… I will cook with meat ball or luncheon meat… wonton and dumpling, which are also easy-to-cook.” (A3, F, 14)

10.Eating out in restaurants or takeaway food

Prevalence of unhealthy options—Some adolescents ate out or consumed takeaway food for lunch with their classmates after school. They usually selected meals from local fast food outlets near their school or home, where unhealthy options dominated.

“Go to buy some takeaway… after school… for lunch…. [Usually] buy McDonald’s… Sometimes buy the street food in Kwai Fong.” (A14, F, 13)

Infrequent but unhealthy options for family meals—Eating out in restaurants with family or friends was infrequent, taking place less than once a week for some adolescents, especially among those with healthy eating habits, regardless of their household income. Some reported buying takeaway food to eat with their family at least once a week. Parents usually made the food choice based on the taste preference of their children, who tended to choose dishes that were more heavily flavored and meat-oriented.

“Will pick those [dishes] less likely eat at home [when eating out]… like spiced salt, deep-fried… [Children] like fried food.” (P4, F, 56)

11.FV consumption

Eating FV once a day at home—Most adolescents had an insufficient consumption of only one serving of fruit and one bowl of vegetables, and this usually limited to one occasion daily. Fruit consumption tended to be a family practice, as the adolescents rarely ate fruit on their own. Most parents would actively prepare ready-to-eat fruit by peeling and cutting fruit into bite-size pieces after family meals at night; some adolescents did not consume fruit every day.

“Will not eat fruit if [son] is too full after eating many vegetables and drinking soup [for dinner], but will usually eat [one a day].” (P2, F, 55)

Adolescents mainly consumed vegetables at home, where dinner was often the only occasion when they ate vegetables. The practice of eating raw vegetables such as salad or sliced cucumber was uncommon. They did not prefer ordering vegetables when eating out due to the perception of them being less value for money, having low availability, and using unhealthy cooking methods.

“Difficult [to order vegetables for takeaway], because those vegetables out there are very expensive… And I do not know how much oil is added. I am afraid every single piece of vegetables is oily.” (A12, F, 18)

Self-serving of fruit—A few adolescents had the habit of serving themselves fruit, especially apples, grapes, and bananas, which can be easily consumed without peeling. This practice was in addition to consuming after-dinner fruit and promoted a more sufficient daily intake.

“Will never cut [fruit] like those oranges. For apple or pear this type, will be lazier and eat the whole after washing.” (A24, F, 13)

Variety of FV at more occasions—Adolescents with sufficient FV intake were more likely to consume a variety of FV and on more occasions every day.

“[Adolescents] will eat oranges, apples, can eat… at least three types of fruit daily, sometimes eating banana. Eating some in the afternoon, and half an hour or an hour after dinner.” (P11, F, 44)

12.Snacking

Varied habit of snacking—Most adolescents said that they did not buy or eat snacks frequently. They commonly ate snacks when they had an emotional trigger or experienced hunger. Some adolescents tried to limit their snacking to a minimum, while some had a habit of snacking that varied from daily to several times per week. For example:

“I love eating curry fishballs… once in 2–3 days… Drinking 1–2 packs [of orange juice] daily.” (A6, F, 14)

Unhealthy snacks at home—Adolescents mostly ate snacks that were available at home and purchased by their parents, and most of these snacks were unhealthy items, such as confectionary, ice cream, and street food.

“Between lunch and dinner, sometimes will eat tofu pudding, egg tart, biscuit, depends on what is available… Sometimes [mum] goes out for grocery shopping, and buy these as well.” (A5, M, 15)

Serving healthy snacks—Only few adolescents had healthy snacking practices of consuming homemade drinks, fruit, and dairy products between meals.

## 4. Discussion

### 4.1. Major Findings

This study identified some key KAP enablers and barriers to healthy eating among adolescents (Table 6).

The adolescents in this study had general knowledge of healthy eating by food categories, unhealthy food choices, and cooking methods. They perceived healthy eating as important, which was consistent with the results of previous studies [13,16], with short-term health outcomes being the underlying drivers of their choices. While having a good body shape was a common reason for them to perceive a healthy diet as important, adolescents who had experienced positive health outcomes in terms of physical fitness, skin condition, and bowel movement because of good dietary habits were more likely to perceive the importance of healthy eating. This positive attitude also promoted their preference for health and improved their willingness to change. They tended to eat meals at home that were prepared by their parents. It was encouraging although unexpected to find that these adolescents had a low frequency of eating out and snacking compared to their counterparts in other Asian countries [24,36,37]. The lower household incomes of our subjects may explain this difference, as the families might have lower budgets for eating out in restaurants and buying snacks.

We found several key KAP gaps that were unfavorable to healthy eating habits among the adolescents. There was a lack of knowledge on the recommended quantities of specific food types that should be consumed, the long-term health outcomes of unhealthy eating, and choosing healthy options for restaurant food and snacks. The attitude gaps included low perceived susceptibility to health problems and a preference for taste and convenience over health. The practice gaps mainly reflected the limited responsibilities of the adolescents in meal preparation and their limited occasions of FV intake.

#### 4.1.1. Knowledge Gaps

The adolescents had limited knowledge of the recommended daily intake of FV and the allowances of salt and sugar. Previous studies have found that adults also have poor knowledge regarding FV portion size [38], and only 28.3% of the Chinese adult participants knew the recommended salt intake [39]. This gap may reflect the omission of such information in most textbooks and in public health education offered in the community. This knowledge gap could lead to adolescents’ having wrong perceptions about meeting dietary recommendations and result in them acquiring NCD despite various public health measures implemented to promote healthy eating [40].

Adolescents rarely acknowledged the serious long-term diet-related health outcomes of unhealthy eating habits, believing that the risk of cardiovascular diseases only applied to older adults. They were not aware that increased metabolic risk, such as obesity and blood pressure, could be cumulative, in that adolescent unhealthy eating habits could lead to an early onset of chronic diseases [11,41].

The adolescents had limited knowledge of making choices regarding healthy snacks. Apart from fruit, yogurt, and milk, other healthy snacks such as nuts, corn, and raw vegetables were seldom mentioned, probably because these are uncommon components of the Chinese diet. Salty and sugary snacks are the major snack items consumed worldwide [42], particularly in children and adolescents [24,43]. The adolescents thought that eating out must be unhealthy, which might be a result of the lack of healthy menus in restaurants in their community [44] or due to them not knowing how to choose the healthier alternatives available in restaurants [45].

#### 4.1.2. Attitude Gaps

Many adolescents with fewer healthy eating habits perceived that the potential harm caused by eating an unhealthy diet was not relevant to them. This was consistent with a study conducted in Hungary that also found that adolescents had poor perceptions of health risk [46]. Similar to the literature on smoking cessation, the absence of immediate health outcomes results in an underestimation of risk and vulnerability [47,48,49]. The emphasis on immediate and visible health benefits of healthy eating, such as skin condition, muscle strengthening for physical fitness and sports performance, and gut health, may be more effective in motivating adolescents to adopt healthy eating habits.

We found that most adolescents did not consider health as the top priority when making their food choices. Food preference is influenced by taste, impression, convenience, and cost [50]. Healthy eating requires adolescents to consider health against these competing considerations, but the desire for immediate gratification over later potential benefits is a common human behavior, especially among adolescents [51]. Adolescents may lack the ability or willingness to prioritize healthy eating over taste and convenience, especially those from low-income families, who tend to be less likely to discuss nutrition-related topics [52].

#### 4.1.3. Practice Gaps

Adolescents had limited responsibility in meal planning, from grocery shopping to cooking, which implies they lack the skills necessary for home meal preparation when their parents cannot cook for them. Parents who need to work may be unable to prepare home meals for lunch or even dinner, leading to adolescents eating out or consuming takeaway food more frequently. A review found that child involvement in meal preparation increased their preference for vegetables and their FV consumption [53]. Giving adolescents the responsibility and opportunity to take part in meal preparation could potentially empower adolescents to prepare healthy meals for themselves and increase their FV intake.

Adolescents obtained most of their FV from home because they rarely chose FV when they aet out in restaurants, which is consistent with the findings of an earlier local study [54]. A major limitation to eating FV was the practice of predominantly eating FV during or after dinner. This is partly related to the Chinese cultural practice of serving fruit at the end of a meal and partly due to the inconvenience of preparation. Another concerning finding was that most adolescents depended on their parents to prepare FV for them. Most adolescents said they did not know how to cook vegetables, and some did not even know how to peel fruit. Ready-to-eat FV (prepared by parents) is often not available when parents are working during the daytime. It is difficult to meet the daily recommended portions in one single occasion of eating unless adolescents learn how to prepare FV for themselves.

### 4.2. Implications of Findings

Adolescents generally perceive themselves as capable of eating healthily because they have learned basic knowledge at school, but their lack of knowledge on the specific dietary recommendations, the lack of observable metabolic effects, and lack of access to healthy choices are concerning in the context of the public health crisis of increasing adolescent obesity and NCD risk.

These findings suggest four major strategies to promote healthy eating among adolescents. The first is to reinforce the knowledge of the specific quantity of daily servings and allowances of various food types. These recommendations should be illustrated in practical terms, such as stating that one fistful of fruit equals one serving and one serving of soft drink constitutes 4/5 of the daily allowance of sugar. This approach is essential to enable adolescents to operationalize the recommendations and follow them more easily.

The second strategy is to help adolescents see the personal relevance of healthy eating. Promoting short-term health outcomes that adolescents can experience immediately, such as improvements in skin condition, physical fitness, sports performance, muscle strength, and sleep quality [55], may induce a more positive attitude towards healthy eating. On the other hand, parents and adolescents need more education on the long-term risk of chronic diseases from the cumulative effect of unhealthy eating that begins from childhood and adolescence.

The third strategy is to provide guidance on healthy food choices in different settings. Adolescents need practical advice on how to balance health concerns against other competing interests when choosing food. This could include reading nutrition labels to choose the snack with the lowest sugar and salt content, ordering burgers with less sauce, and buying healthy drinks in family-size portions at a lower unit cost to save money.

The fourth strategy is to enable FV consumption on more occasions. To eat FV during breakfast and lunch and as snacks, parents should involve their adolescents in food preparation to empower them to proactively select and consume FV and healthy meals without depending on their parents.

### 4.3. Strengths and Limitations

A strength of this study was its identification of key KAP gaps and strategies to inform future interventions by health education providers. The wide diversity of subject backgrounds with regard to age, gender, and FV consumption enhanced the representativeness and validity of the findings.

Another strength was the involvement of parent and adolescent dyads in the interviews. As parents were responsible for the preparation of most meals and purchasing food, they were able to provide Appendix A on the KAP of their adolescents. They could also cross-validate their adolescent’s perspectives on healthy eating. For example, one parent could further explain that her son’s habit of eating more chicken and eggs was for the purpose of muscle strengthening. Another supplied the information that her daughter remembered the harms of unhealthy eating as she had experienced the consequences of eating oily food. The information provided by the parents could also be confirmed by the adolescents and vice versa to provide clarification and minimize misunderstandings.

This study has three potential limitations. First, recall bias and response bias existed, as the research findings relied on self-reported information and participants might have provided socially desirable responses; the dyadic interview approach and non-sensitive subject matter might have minimized these biases. Second, all the parental subjects were mothers, so the opinions of fathers were lacking. However, as mothers play a dominant role in household affairs and food preparation, particularly in Chinese culture, they are usually more familiar with their adolescent’s healthy eating KAP. Third, the subjects were recruited from families of the TFES and a comparative cohort of low-income families; thus, they may not be fully representative of all adolescents in the population, though this was not the intention of this qualitative study. For generalizability, further research using quantitative methods on a larger and more dispersed sample with a more diverse socioeconomic background should be carried out.

## 5. Conclusions

Among adolescents from low-income families in Hong Kong, the key KAP gaps regarding healthy eating were: insufficient knowledge of recommended daily servings or allowances of specific food types; perceived low susceptibility to acquiring NCD due to unhealthy eating; unhealthy snacking and food choices in restaurants; and insufficient FV intake. Initiatives aiming to promote healthy eating should illustrate the quantity of daily servings in practical terms (e.g., fistful of fruit); promote short-term, tangible benefits which adolescents can experience immediately (e.g., skin condition, sports performance); offer practical advice on making healthy food choices in different realistic settings (e.g., ordering burgers with less sauce); encourage more frequent FV consumption, including at breakfast and lunch and as snacks; and encourage adolescent involvement in food preparation. These steps can empower adolescents to acquire healthy eating habits that they can sustain into adulthood.

## Figures and Tables

**Figure 1 nutrients-14-02857-f001:**
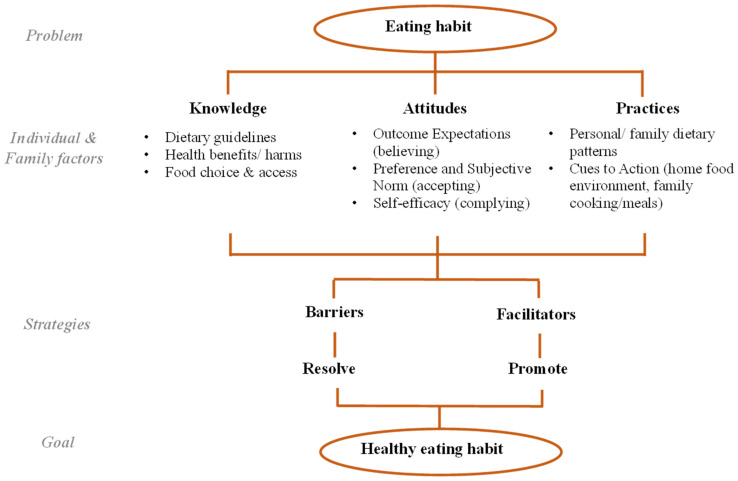
Theoretical framework for family interview: knowledge, attitudes, and practices model.

**Table 1 nutrients-14-02857-t001:** Distribution of subject characteristics.

Characteristics			Participants (%)	
Adolescents	Gender	Female	13	(52%)
		Male	12	(48%)
	Age (years)	Mean ± SD	14.84 ± 2.08	
		12–13	8	(32%)
		14–16	10	(40%)
		17–19	7	(28%)
	FV intake per day	Mean ± SD	3.6 ± 1.53	
		≥5 servings (healthy)	8	(32%)
		3–4 servings (average)	11	(44%)
		1–2 servings (unhealthy)	6	(24%)
Parents	Gender	Female	25	(100%)
	Age (years)	Mean ± SD	49.24 ± 4.65	
		40–49	14	(56%)
		50–59	11	(44%)
	Participation in nutrition workshop	Yes	10	(40%)
Household	Monthly income	Median	HK$13,500–19,999	
		>HK$27,000	4	(16%)
		HK$20,000–26,999	5	(20%)
		HK$13,500–19,999	8	(32%)
		<HK$13,500	8	(32%)

FV = fruit and vegetables, SD = standard deviation.

**Table 2 nutrients-14-02857-t002:** Characteristics of parent–adolescent dyads by family.

Family	Characteristics of		Household Income	Participation in Nutrition Workshop	Eating Status by FV Intake
	Adolescents	Parents			
1	F, 18 y	F, 51 y	0.75–1 median		Healthy
2	M, 16 y	F, 55 y	Below 0.5 median		Average
3	F, 14 y	F, 49 y	0.5–0.75 median	Participated	Average
4	M, 14 y	F, 56 y	Below 0.5 median	Participated	Average
5	M, 15 y	F, 55 y	0.75–1 median	Participated	Unhealthy
6	F, 14 y	F, 42 y	0.5–0.75 median		Unhealthy
7	F, 14 y	F, 51 y	0.75–1 median		Healthy
8	M, 14 y	F, 51 y	Below 0.5 median	Participated	Average
9	F, 17 y	F, 40 y	Above median	Participated	Healthy
10	F, 14 y	F, 48 y	0.5–0.75 median		Average
11	F, 13 y	F, 44 y	Below 0.5 median		Healthy
12	F, 18 y	F, 47 y	0.5–0.75 median		Unhealthy
13	M, 13 y	F, 48 y	Below 0.5 median	Participated	Unhealthy
14	F, 13 y	F, 49 y	Below 0.5 median	Participated	Unhealthy
15	M, 13 y	F, 46 y	0.75–1 median		Healthy
16	M, 12 y	F, 48 y	Below 0.5 median	Participated	Average
17	M, 15 y	F, 52 y	Below 0.5 median	Participated	Healthy
18	M, 19 y	F, 55 y	0.5–0.75 median		Average
19	F, 12 y	F, 55 y	0.75–1 median	Participated	Unhealthy
20	F, 17 y	F, 55 y	Above median		Healthy
21	M, 15 y	F, 47 y	Above median		Average
22	M, 13 y	F, 52 y	0.5–0.75 median		Healthy
23	F, 17 y	F, 48 y	Above median		Average
24	F, 13 y	F, 40 y	0.5–0.75 median		Average
25	M, 18 y	F, 47 y	0.5–0.75 median		Average

F = female; M = male, y = years of age.

**Table 3 nutrients-14-02857-t003:** Summary of themes in adolescent knowledge of healthy eating.

Themes	Knowledge in Common	Knowledge Gap	
		Insufficiency	Inaccuracy
Dietary recommendations	Relative portions of food categories e.g., food pyramid; low sugar, oil, salt	Recommended daily servings or allowance e.g., lunchbox 3-2-1 portion; salt content in noodle soup	Underestimating the recommended servings e.g., one apple daily
Health outcomes of healthy eating	Observable short-term outcomes e.g., constipation, body weight, skincare, sore throat	Specific benefits of eating FV e.g., relieved body pain, skincare, detoxification Long-term outcomes e.g., CVD, cancer	
Nutrition content of food	Food sources of fat, salt, sugar, and nutrients e.g., protein, calcium Interpretation of nutrition label or claim	Healthy snack options e.g., yogurt, fruit, nuts	
Access to healthy meals	Unhealthy cooking methods used for restaurant and takeaway food	Healthy cooking methods e.g., boiling, steaming Ways to identify healthier restaurant meals	

**Table 4 nutrients-14-02857-t004:** Summary of themes on adolescent attitudes towards healthy eating.

Themes	Attitudes in Common	Attitude Gap	
		Insufficiency	Inaccuracy
Outcome expectations for healthy eating	Being necessary for weight control Experience of negative outcomes from unhealthy eating habits	Experience of positive outcomes from healthy eating habits	Healthy eating only being necessary for older adults and not for young people
Food preferences	Taste preference for unhealthy food	Prioritizing health among other preferences e.g., taste, convenience	Perceived inferior taste of healthy food
Self-efficacy regarding adopt healthy eating	Assessing health by body shape	Strategies to eat healthily with friends or on their own Skills of food preparation	

**Table 5 nutrients-14-02857-t005:** Summary of themes on adolescent practices of healthy eating.

Themes	Practices in Common	Practice Gap	
		Insufficiency	Unhealthy
Grocery shopping for healthy food	Not being in the habit of reading nutrition labels or health claims	Reading nutrition labels to determine healthy alternatives Accompanying parents for grocery shopping	
Eating home-prepared meals	Parents preparing meals Eating ready-to-eat or easy-to-cook foods for breakfast Eating grains, vegetables, and meat for lunch and/or dinner		Adolescent’ use of unhealthy ingredients when cooking for themselves
Eating out in restaurants or consuming takeaway food	Eating out or consuming takeaway food for lunch after school	Infrequent eating out with family or friends	Availability of unhealthy eating out options Occasionally buying unhealthy takeaway food for family meals
FV consumption	Eating FV once a day at home Parents preparing ready-to-eat fruit	Preparing fruit themselves Eating a variety of FV on more occasions	
Snacking	Infrequently buying and eating snacks Eating snacks available at home	Serving healthy food and homemade drinks as snacks	Unhealthy snacking habits e.g., more than 3 times a week

FV = fruit and vegetables.

**Table 6 nutrients-14-02857-t006:** Summary of key KAP enablers and barriers to healthy eating in adolescents.

Enablers	Barriers
Knowledge	
General knowledge of healthy eating by food categories, unhealthy food choices, and cooking methods	Not knowing recommended quantities of specific food types Unaware of the long-term health outcomes and healthy options of restaurant food and snacks
Attitudes	
Perceived importance of healthy eating for achieving short-term health outcomes Positive experience of health outcomes	Low perceived susceptibility Low preference for health over taste and convenience
Practices	
Eating home-prepared meals Infrequent eating out and snacking	Limited responsibility in meal preparation Limited occasions for FV intake

FV = fruit and vegetables.

## Data Availability

All data generated or analyzed during this study are included in this published article and its Appendix A.

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
