# Peer review of "Adolescent Knowledge, Attitudes and Practices of Healthy Eating: Findings of Qualitative Interviews among Hong Kong Families"

_nutrients, 2022, doi:10.3390/nu14142857_

Round 1
Reviewer 1 Report
The manuscript is interesting and novel. The authors study important aspects of healthy eating. The introduction is adequate, the methodology used is sufficient. The results support the discussion. However, I have the following comments.
I. Major Comments:
1. Considering that they are adolescents and the authors evaluate aspects of healthy eating. In the introduction it is necessary to include a paragraph on food, nutrition and health. For example, chronic non-communicable diseases or anorexia nervosa.
2. The results are properly presented. However, it would be nice to improve the presentation of results, by grouping comments.
3. In the discussion, briefly include aspects of public health and nutrition.
4. I suggest improving the conclusion. It's a bit confusing and very general.
II. Minor comments:
1. Improve the wording of the objective of the study
2. It would be good to include a table with the average information of the subjects studied, for example: age, weight, BMI, etc.
Reviewer 2 Report
The article is very well written, there are some minor technical errors, there should be square brackets when citing literature in the text.
Good job.
Round 2
Reviewer 1 Report
Authors answered all my comments. Therefore, manuscript can be accepted in the present form.